# Tough double network hydrogels with rapid self-reinforcement and low hysteresis based on highly entangled networks

Ruixin Zhu[1], Dandan Zhu[1], Zhen Zheng ®[1] & Xinling Wang ®[1,2] ✉

Most tough hydrogels are reinforced by introducing energy dissipation mechanisms, but simultaneously realizing a high toughness and low hysteresis is challenging because the energy dissipation structure cannot recover rapidly. In this work, high mechanical performance highly entangled double network hydrogels without energy dissipation structure are fabricated, in which physical entanglements act as the primary effective crosslinking in the first network. This sliding entanglement structure allows the hydrogel network to form a highly uniform oriented structure during stretching, resulting in a high tensile strength of ~3 MPa, a fracture energy of 8340 J m$^{-2}$ and a strain-stiffening capability of 47.5 in 90% water content. Moreover, almost 100% reversibility is obtained in this hydrogel via energy storage based on entropy loss. The highly entangled double network structure not only overcomes the typical trade-off between the high toughness and low hysteresis of hydrogels, but more importantly, it provides an insight into the application of entanglement structures in high-performance hydrogels.

So far, PAMPS/PAAm-based double network (DN) hydrogels have a history of 20 years. The fracture energy of this hydrogel can reach 100–1000 J m$^{-2}$ under a 90% water content, and its compression strength up to 40 MPa[1–4]. More significantly, the energy dissipation mechanism of traditional double-network (TDN) hydrogels has inspired the development and application of many high-strength hydrogels. For example, the DN hydrogel with an alginate-Ca$^{2+}$ network as the energy dissipation structure achieved a fracture energy of 9000 J m$^{-2}$ [5], but its toughness declined dramatically after the first stretching cycle and showed high hysteresis. Moreover, this low reversibility is not a exclusive shortcoming of DN hydrogels, but rather an inevitable problem for most hydrogels with energy dissipation mechanisms[5–9]. Instantaneous recovery cannot be achieved even if non-covalent bonds[10–15], nanoparticles[16,17] and crystal structures[18,19] with recoverable energy dissipation mechanisms are used. Reconstruction of damaged structures usually takes minutes or hours which is much longer than the energy dissipated time, and results in hard instantaneous mechanical property recovery at the experimental observation scale. Although some elastic gels that do not use energy

dissipation mechanisms exhibit high recovery, their show low tensile strength and toughness[20–23]. For example, the polyacrylamide hydrogel using hyperbranched silica nanoparticles (HBSPs) as the main junction point showed only 1.3% hysteresis, but its tensile strength was only 150 kPa[23]. Polymerizable rotaxane hydrogels (PR-Gel) had a reversibility of 97.5%, but their tensile strength was only 80 kPa[21]. This is consistent with the fact that elastic gels are known to be brittle and notch-sensitive[24,25]. In other words, the stretchability and strength decrease significantly when a sample contains nicks or other features that produce uneven deformation[5]. Polyrotaxane PEG hydrogels with strain-induced crystallization prepared by Koichi Mayumi and Kohzo Ito et al. achieved a combination of high toughness and low hysteresis to a certain extent. With a water content of 70%, their tensile strength, fracture energy and reversibility were 3 MPa, 2200 J m$^{-2}$ and almost 100% respectively. Nevertheless, the synthesis of the polyrotaxane is complicated[26]. Therefore, how to prepare hydrogels with high toughness and low hysteresis simply and efficiently is still an urgent problem.

Elastic networks without energy dissipation structures can improve the reversibility of hydrogels, and their weak mechanical

[1]School of Chemistry and Chemical Engineering, Shanghai Jiao Tong University, Shanghai 200240, China. [2]State Key Laboratory of Metal Matrix Composites, Shanghai Jiao Tong University, Shanghai 200240, China. ✉e-mail: xlwang@sjtu.edu.cn

properties may be improved by adjusting the network structure. Many studies have shown that the network structure significantly affects the hysteresis[27], elasticity[28], and fracture behavior[29,30] of hydrogels. The classical continuum theory used to predict the properties of hydrogels is usually based on a simplified model that assumes a perfect, homogeneous three-dimensional network[31–33]. However, recent studies have shown that hydrogel networks are spatially inhomogeneous, especially those obtained by radical polymerization[34,35]. Spatial inhomogeneity refers to inhomogeneous crosslinking, usually resulting in a mixture of densely and loosely crosslinked domains in the network with distances on the order of 10–100 nm[36]. Yang et al. reported the effect of non-uniform network structure on the fracture behavior of hydrogels[29]. Gong et al. also discovered submicrometer-scale voids in a double network hydrogel and explored their effect on the yield and necking under uniaxial stretching[37]. Many literatures have demonstrated that a non-uniform microstructure of hydrogels will markedly affects their macroscopic properties, but methods to avoid or reduce this network structure inhomogeneity and thus improve the fracture behavior and hysteresis of the material have not been studied.

In this work, we prepare highly entangled double network (HEDN) hydrogels, in which many physical entanglements act as effective crosslinks for the first network. The sliding entanglements cause HEDN hydrogels to form a highly uniform orientation structure under tensile strain, which is summarized as stretching-induced highly uniform orientation behavior. The prepared hydrogels show high tensile strength, high toughness and strain-stiffening characteristics. Moreover, the external energy applied to the hydrogels is not dissipated, but rather stored in highly oriented molecular chains as an entropy loss, which endows the hydrogels with low hysteresis. Therefore, high toughness and low hysteresis are achieved by a highly entangled network structure.

## Results

### Design and properties of hydrogels with highly entangled double network structure

Highly entangled double-network (HEDN) hydrogels can be regarded as DN hydrogels considering that its network structure is similarity to traditional double-network (TDN) hydrogels. The difference between HEDN hydrogels and TDN hydrogels lies only in the design of the first network. In TDN hydrogels, the first network is usually a brittle polyelectrolyte with dense chemical crosslinks prepared in a dilute monomer solution. In contrast, the first network of HEDN hydrogels is a tough polymer with dense physical entanglement points obtained in a high-concentration monomer solution. As shown in Fig. 1a, HEDN1st hydrogels are produced in high-concentration monomer solutions. AAm is selected to form the polymer backbone, and AMPS is introduced into the polymer backbone in order to increase the osmotic pressure of the polymer network so that the polymer networks can absorb more second network monomers[38]. Minimal chemical crosslinker MBAA (0.001 mol%) is used in HEDN1st network to prevent the disentanglement of physical entanglement points. The successful preparation of HEDN1st hydrogels containing different AMPS components is demonstrated in Fig. 1b. The peak at 1184 cm$^{-1}$ is attributed to the stretching vibration band of S=O bond of sulfonate groups, and the peak at 1035 cm$^{-1}$ result from the stretching vibration absorption of S-O. These peaks intensified upon increasing the AMPS content, indicating the successful incorporation of AMPS into the polymer backbone.

Highly entangled hydrogels mean that there is more entanglement per unit volume under the same solid content. The entanglement degree of HEDN1st hydrogels is controlled by the concentration of prepolymer solutions. Entanglements will significantly affect the equilibrium water content of hydrogels. For example, the water content of HEDN1st hydrogels increases significantly with the increase of

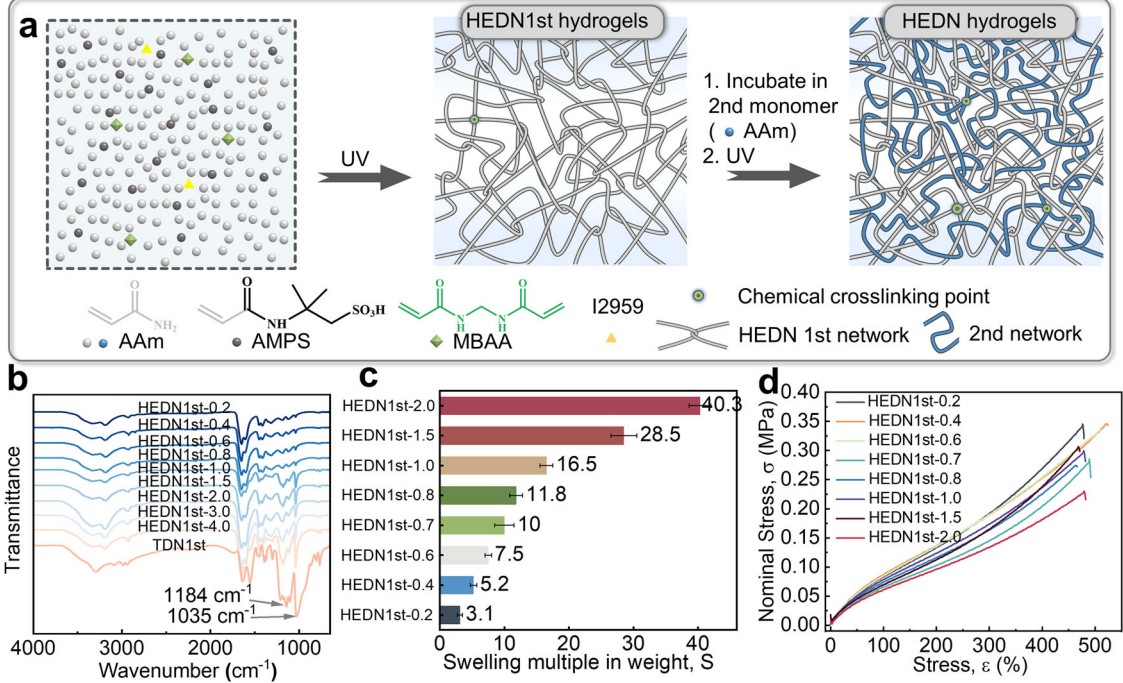

**Fig. 1 | Highly entangled double network design and first network parameter. a** Schematic illustration of the preparation method of the HEDN hydrogels. First, the HEDN1st hydrogels are prepared by copolymerization of AAm and AMPS using ultraviolet (UV) initiated free radical polymerization, MBAA and I2959 are used as crosslinker and initiator, respectively. Then HEDN1st hydrogels are incubated in the second network monomer solution for 24 h. Finally, the second network is generated in situ to obtain HEDN hydrogels. **b** FTIR spectra of HEDN1st xerogels with different AMPS contents. **c** Swelling multiple in weight of HEDN1st hydrogels with different AMPS contents in second network monomer solution. Error bars represent mean +/− standard deviation (n = 5). **d** Tensile stress-strain curves of HEDN1st hydrogels, water content: 89%.

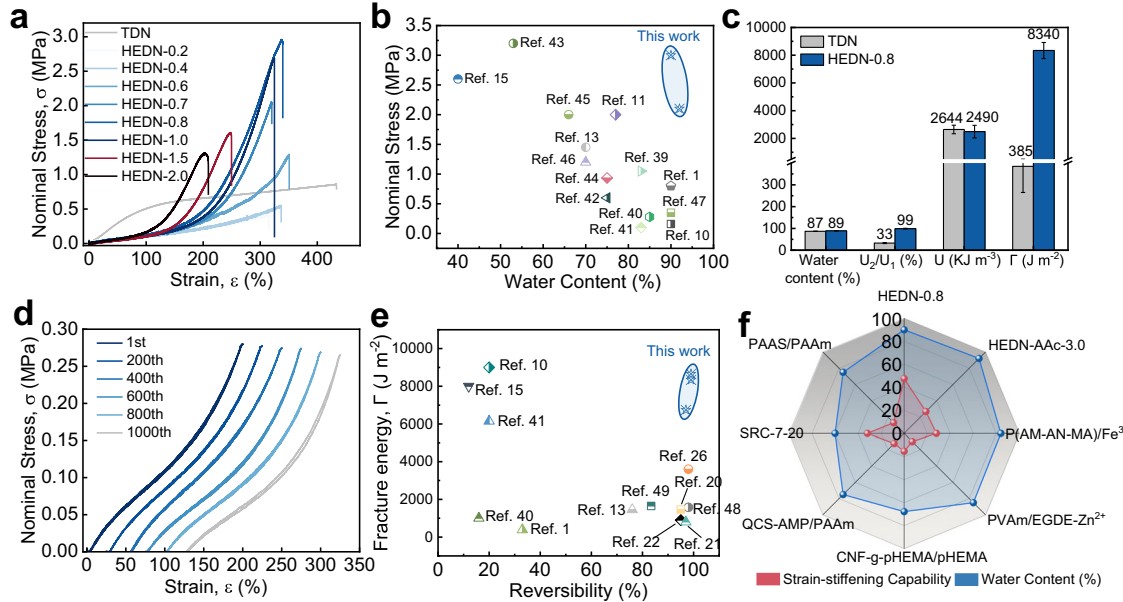

**Fig. 2 | Mechanical properties of HEDN hydrogels. a** Tensile stress–strain curves of TDN hydrogel and HEDN hydrogels with different AMPS contents. Note: Preparation parameters of TDN hydrogel: 1st AMPS (1 M) MBAA (4 mol%), 2nd AAm (4 M) MBAA (0.05 mol%). **b** Ashby diagrams of nominal stress versus water content of previously-reported DN hydrogels. **c** Comparison of water content, reversibility ($U_2/U_1$), toughness (U), and fracture energy (Γ) between TDN hydrogels and HEDN-0.8 hydrogel. Error bars represent mean +/− standard deviation (*n* = 5). **d** One thousand consecutive loading-unloading cycles for the HEDN-0.8 hydrogel at 200% strain. Shifts are applied to the 200th, 400th, 600th, 800th, and 1000th cycle curves. **e** Ashby diagrams of fracture energy versus reversibility in previously-reported hydrogels. Hydrogels with very high toughness but very low reversibility were excluded. **f** Radar charts showing the strain-stiffening capability and water content parameters across hydrogels with strain-stiffening behavior (details in Supplementary Table 3).

W, eventually reaching equilibrium at ~99.7%, while the water content is 95.4% when W is 2 (Supplementary Fig. 1). The swelling rate also shows a similar phenomenon (Supplementary Fig. 2). Many entanglements effectively inhibit the swelling of networks. The entanglements also increase the stiffness of hydrogels[9,20]. To eliminate the effect of osmotic pressure on stiffness, we prepared hydrogels swollen by salt solutions (3 M NaCl). As shown in Supplementary Fig. 3, there is no significant difference in the stress-strain curves and stiffness of hydrogels swollen by water or salt solution, whether at low AMPS content, such as HEDN1st-0.2, or at higher AMPS content, such as HEDN1st-2.0, which means that the elasticity caused by osmotic pressure is negligible in this system. The HEDN1st hydrogels obtained at five different concentrations were processed into samples with a water content of 89% to eliminate the influence of polymer contents on stiffness. Therefore, the stiffness of the gel is determined by the degree of entanglement in this work. As shown in Supplementary Fig. 4 (pink area), the stiffness of HEDN1st increases significantly as the W value decreased, indicating a significant increase in physical entanglements. The tensile strength also increases with the increase of entanglement, but the elongation at break is opposite (Supplementary Fig. 5). Therefore, HEDN1st prepared at the highest monomer concentration (W2) are regarded as highly entangled hydrogels. Then the swelling degree of HEDN1st hydrogels in the monomer solution of the second network is investigated because it would affect the ratio of the first network and the second network of HEDN hydrogels, leading to drastic differences in mechanical properties. As shown in Fig. 1c, HEDN1st-0.2 hydrogel can only absorb a little more than twice its own weight when placed in the second network monomer solution due to the weak charge repulsion. Upon increasing the concentration of the charged monomer AMPS, HEDN1st hydrogels absorb more monomer solution, and HEDN1st-2.0 hydrogel absorbs close to 40 times its own weight without breaking. Thus, the results in Supplementary Fig. 6 show that the HEDN1st-0.2 hydrogel has the lowest water content of 90.5%, while the HEDN1st-2.0 hydrogel has the highest (98.5%). Figure 1d shows the tensile stress-strain curves of HEDN1st hydrogels at the same water

content. The tensile strength of HEDN1st hydrogels decreases slightly with the increase of AMPS. Since the concentration of the prepolymer solution does not change, i.e., the degree of entanglement is the same, the elongation at break and initial modulus are almost unchanged.

As shown in Fig. 1a, HEDN1st hydrogels were immersed in the second network monomer solution and allowed fully swell. Then the second network was formed in situ under UV irradiation, and finally immersed in deionized water to fully swell to obtain HEDN hydrogels. These hydrogels have a same high transparency as TDN hydrogels (Supplementary Fig. 7). Figure 2a compares the uniaxial tensile curves of HEDN hydrogels and a TDN hydrogel. HEDN hydrogels exhibit typical strain-stiffening behavior, and its fracture stress reach up to 3 MPa. The compressive strength of HEDN hydrogels is also comparable to that of TDN hydrogel, its highest compressive strength reaches 48 MPa (Supplementary Fig. 8). The fracture strength of the HEDN hydrogels first increases and then decreases, and the fracture strain gradually decreases with the increased AMPS because there were different proportions of the first and second networks. The HEDN1st-0.2 hydrogel shows the lowest swelling rate, which increases with the content of AMPS, leading to a gradual increase in the proportion of the second network. HEDN-0.8 hydrogel has the most suitable ratio of the first network and second network, and the synergistic effect based on the two networks provides it with the highest breaking strength. The role played by the first network gradually decreases with the further increase of AMPS, and the larger swelling rate makes the hidden chain length of the first network stretch more, so the elongation at the break of HEDN hydrogels begins descending.

It is worth noting that HEDN-0.8 hydrogel has a water content close to 90% (Fig. 2c). The mechanical strength of hydrogels and their water content are typically inversely correlated, making it difficult to achieve high mechanical properties at high water contents. Compared with almost all previously-reported DN hydrogel, the HEDN hydrogels here show better mechanical properties at high water contents (Fig. 2b)[39–47]. Not only that, compared with the TDN hydrogel, the HEDN gels show good stretch reversibility and fatigue resistance. The

high toughness of the TDN hydrogel originates from the large amount of energy absorbed by covalent bond breakage in the brittle first network. However, the broken covalent bonds cannot be repaired, making the TDN hydrogel weak after the first stretching. The large hysteresis loops of the TDN hydrogel in Supplementary Fig. 9 indicate that many covalent bonds in the first network are broken to dissipate energy. The second stress-strain curve of the TDN hydrogel is very different from the first stretching cycle, and the reversibility $U_2/U_1$ at 200% strain is only 33%, while the reversibility of the HEDN hydrogels reach 99% (Fig. 2c and Supplementary Fig. 10). Moreover, HEDN-0.8 hydrogel can withstand 1000 stretching cycles with no deterioration at 200% strain (Fig. 2d). The lower reversibility of HEDN hydrogels at 300% strain is attributed to partial polymer chain fracture (Supplementary Fig. 11). TDN and HEDN-0.8 hydrogels show almost equal toughness values of 2644 kJ m$^{-3}$ and 2490 kJ m$^{-3}$, respectively (Fig. 2c). However, the fracture energy of the two hydrogels is 385 J m$^{-2}$ and 8340 J m$^{-2}$, respectively, which are more than 20 times different. A high fracture energy and high reversibility also is another typical trade-off of hydrogels, because a high fracture energy usually involves an energy dissipation mechanism, and the recovery of energy dissipation structures is slow. However, HEDN hydrogels have higher fracture energy and reversibility than most hydrogels (Fig. 2e)[48,49]. Further analyzing and differentiating the tensile curves of TDN and HEDN hydrogels (Supplementary Fig. 12), shows that the modulus of the TDN hydrogel gradually decreases to a constant value with the increase of strain, while the modulus of HEDN hydrogels increases rapidly from a constant value to a larger value, showing typical strain-stiffening behavior. The strain-stiffening capacity of the HEDN-0.8 hydrogel reach up to 47.5 times (Supplementary Fig. 13), which is higher than almost previously-reported hydrogels with strain-stiffening (Fig. 2f). The higher strain-stiffening capacity allows soft materials to harden more rapidly under large deformation, which is defined as rapid self-reinforcement behavior.

The good mechanical properties of the HEDN hydrogels come from the synergistic effect of the two networks. The degree of entanglement of the first network, the ratio of the first network to second network, the entanglement degree of the second network and amount of chemical crosslinker will affect the synergistic effect. As shown in Figure S4 (pink area), the HEDN hydrogels prepared from HEDN1st hydrogels with higher entanglement have higher stiffness, indicating higher entanglement. Thus, HEDN-0.8-W2 hydrogel has the highest strength (Supplementary Fig. 14). Due to the constant total monomer concentration, the AMPS barely affect the entanglement degree of HEDN1st hydrogels (blue area in Supplementary Fig. 4), but the swelling rate of HEDN1st hydrogels could be controlled by changing the content of AMPS, thus adjusting the proportion of the two networks in HEDN hydrogel. The proportion of the second network increases with the increase of AMPS. It is worth noting that the increase of the second network does not increase the entanglement degree of HEDN hydrogels (blue area in Supplementary Fig. 4), because the HEDN1st network will swell with the increase of AMPS, which reduces the number of entanglements per unit volume, but the introduction of the second network will increase entanglements. Therefore, the total amount of entanglement in the HEDN hydrogels is controlled by both effects. The great difference in tensile properties depicted in Fig. 2a indicates that HEDN-0.8 hydrogel has the optimal network ratio.

Increasing the concentration of the second monomer solution will significantly enhance the total entanglement of HEDN hydrogels (purple area in Supplementary Fig. 4), thus affecting its mechanical properties. As shown in Supplementary Fig. 16, the tensile strength of HEDN hydrogels first increases and then decreases with the increase of entanglement, which means that the more entanglements are not the better. Within a reasonable range, the increase of chemical crosslinking agents does not affect the stiffness of the hydrogels (green area in Fig. S4), which is attributed to the fact that the number of entanglement points is much higher than that of chemical cross-linking, which plays a dominant role in the effect of stiffness. This is consistent with what Suo et al. reported[20]. However, the mechanical properties of HEDN hydrogels first increased and then decreased with the increase of chemical crosslinkers (Supplementary Fig. 15), indicating that the appropriate dosage of chemical crosslinkers is also an important factor in achieving high performance of HEDN hydrogels. Finally, we should explain the necessity of double networks. It can be seen from Supplementary Fig. 17a that the tensile strength of HEDN-0.8 is much higher than that of HEDN1st-0.8 under the same water content. However, due to the introduction of a second network, the entanglement degree of HEDN-0.8 is higher than that of HEDN1st-0.8. In order to exclude this effect, HEDN-0.8-W4 with the same entanglement degree as HEDN1st-0.8 is used for comparison. Supplementary Fig. 17b shows that the mechanical strength of HEDN hydrogels with double networks is still much higher than that of HEDN1st with single networks.

In conclusion, HEDN hydrogels with appropriate parameters achieve the optimal synergistic effect of two independent cross-linked networks. The fundamental reason is that sliding entanglement causes the polymer network to form a highly uniform orientation structure under tensile strain. Therefore, the following work will compare the structural changes of HEDN hydrogels composed of sliding entanglements and TDN hydrogels composed of fixed chemical crosslinking before and after stretching, and further explain the toughening mechanism of HEDN hydrogels.

### Self-reinforcement, high reversibility mechanism of highly entangled double network structures

The TDN hydrogel is not a completely uniform network[37]. As shown in Fig. 3a, c, the pores of the freeze-dried TDN hydrogel are unevenly distributed. This uneven pore distribution is not a random distribution of individual pores but rather large and small pores aggregate in different regions or microdomains. The areas in the orange dotted circle have significantly larger pore diameters than in blue areas. According to statistical analysis, the average diameter of large pores is 5.26 ± 1.07 μm, and the average diameter of small pores is 3.24 ± 0.88 μm (Supplementary Fig. 18a–c). These pores were produced by defects (or large voids) in the first network during free-radical polymerization, and the high-density chemical crosslinking ensures the stability of these defects. Thus, the second network formed in the defects can swell more freely. The large anomalous scattering in the TDN hydrogel's scattering profile in Fig. 4d also indicates the presence of microdomains. In contrast, HEDN hydrogels show relatively weak shoulder peak, indicating that the uniformity of the network has been improved to some extent. The more uniform network distribution shown in Fig. 3e and Supplementary Fig. 18d–f also demonstrates this. The roughness analysis of both hydrogels in Supplementary Fig. 19 also shows a similar phenomenon. The root mean square roughness Rq of the TDN hydrogel is 133.2 nm, while the Rq of the HEDN-0.8 hydrogel is only 27.4 nm. The high roughness of the gel is attributed to the different swelling degrees of different microdomains. A more uniform network can effectively avoid the stress concentration within the hydrogels[29].

The brittle first network of the TDN hydrogel begins to break when stretched, and the polymer chains (those chains with small hidden lengths) in the second network are selectively elongated orientation (Fig. 3d). The first network is eventually broken completely at high strain, and the remaining strain is mainly provided by the flexible second network. Therefore, the TDN hydrogel has a higher initial modulus, but the modulus decreases upon increasing the strain. Whereas for the HEDN hydrogels, the entanglement points in the first network begin to slide under stress and orient the polymer chains along the direction of the applied force, while also incorporating the entangled second network into it (Fig. 3h). The sliding on the

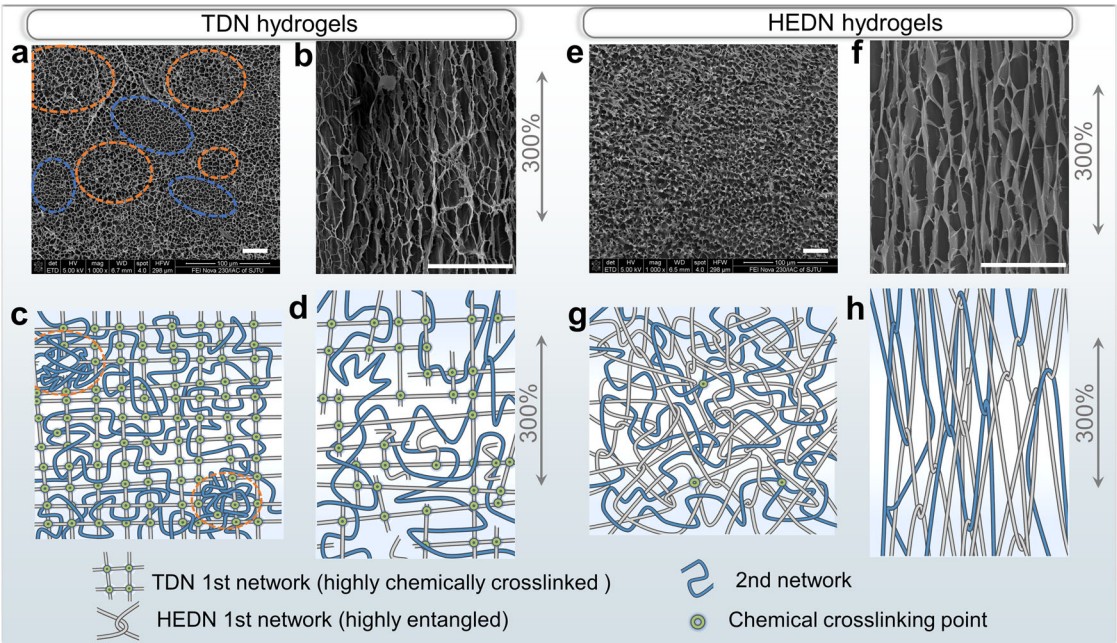

**Fig. 3 | Structural changes of hydrogels before and after stretching.**
**a** Microstructure of unstretched freeze-dried TDN hydrogel. Larger or smaller void structures can be observed at the orange or blue circle marks. **b** Microstructure of freeze-dried TDN hydrogel stretched to 300% of its original length. Diagram of structural changes in the TDN hydrogel before (**c**) and after (**d**) stretching.
**e** Microstructure of unstretched freeze-dried HEDN-0.8 hydrogel. Note: This image indicates a more uniform network compared to Fig. 3a, but there are still microdomains that only can be detected by SAXS. **f** Microstructure of freeze-dried HEDN-0.8 hydrogel stretched to 300% of its original length. Diagram of structural changes of the HEDN-0.8 hydrogel before (**g**) and after (**h**) stretching. Scale bars in SEM images: 30 µm.

entanglement points homogenizes the polymer chains with different hidden lengths, i.e., all polymer chains are equally stretched. The HEDN hydrogels show almost no polymer chains breakage during the initial stretching, and only polymer chain orientation occurs. In this process, energy is stored as an entropy loss. When the external force is removed, this energy will be released in the form of an entropy increase. When the HEDN hydrogels are stretched to a certain extent, the polymer chains are highly oriented, and the material becomes very rigid in the direction of the applied force, thus exhibiting typical strain-stiffening behavior. As shown in Supplementary Fig. 12b, the modulus of HEDN hydrogels continues to increase with strain. The strain-stiffening capacity of HEDN-0.8 and HEDN-1.0 exceeds 40 (Supplementary Fig. 13), showing strong self-reinforced behavior, which is conducive to avoiding material damage under large strains.

Birefringence is often used to determine the orientation of polymer chain segments because both birefringence and stress are proportional to the end-to-end distance of a chain when the orientation of the chain segments follows a Gaussian distribution[50,51]. The polymer chain orientation of the TDN hydrogel and HEDN-0.8 hydrogel are observed by a polarized light microscope, as shown in Fig. 4a, b. Above 100% strain, the interference color of both hydrogels gradually increases with the strain. However, when the HEDN-0.8 hydrogel is stretched from 300% to 350% strain, the interference color changes abruptly, which is generated by the high-level orientation of polymer chains caused by high-density chain entanglements. This is also quantitatively demonstrated by the change in birefringence. As shown in Fig. 4c, although the birefringence of both hydrogels increases with the strain, the TDN hydrogel show a gentler increase, while the birefringence ($\Delta n$) of the HEDN-0.8 hydrogel jumps from $7.5 \times 10^{-4}$ at 250% strain to $32.1 \times 10^{-4}$ at 350% strain. This strain range exactly corresponds exactly to the abrupt change in the stress-strain curve. Supplementary Movies 1 and 2 more intuitively show the in-situ stretching interference color change trend of the two hydrogels under polarized mode and are in good agreement with the Michel–Levy chart[52]. Strain-

stiffening behavior is caused by the higher orientation of polymer chains under large strains.

Because the hydrogel network has inhomogeneous cross-linked microdomains, the deformation of these microdomains is different when stretched, so the orientation of the network is not completely uniform. If when the network is stretched and the microdomains disappear, the resulting oriented structure is regarded as highly uniformly oriented. Birefringence can be used to clearly compare the orientation degree of the polymer chains of both hydrogels during stretching, but the uniformity of the orientation structure at the sub-micron scale is unknown. The orientation uniformity can be visually compared through the SEM images of TDN hydrogel (Fig. 3b) and HEDN hydrogels (Fig. 3f). Obviously, the stretched HEDN hydrogels have a more orderly orientation. SAXS (Fig. 4d) also shows that both the undeformed TDN and HEDN hydrogels have obvious shoulder peaks, indicating that both have heterogeneous structures, which are caused by ununiform chemical crosslinking and ununiform polymer entanglement, respectively. The scattering profile of the TDN hydrogel stretched to 300% still retain strong anomalous scattering, indicating that stretching does not affect its microdomains. In contrast, the scattering profile of the HEDN-0.8 hydrogel stretched to 300% has almost no anomalous scattering. This indicates that although there is a certain non-uniform entanglement network in the undeformed HEDN hydrogels, the sliding entanglements make the stretched polymer network achieve uniform orientation. The 2D SAXS images of both hydrogels at different stretching ratios are shown in Supplementary Fig. 20. As the tensile strain increases, the scattering patterns of both hydrogels change from isotropic to striped, but for different reasons. For TDN hydrogel, the microdomain structure transition from isotropic to an anisotropic ellipsoid, whereas for the HEDN-0.8 hydrogel, this occurred due to the high order of the polymer chains. To study change in the microdomain structure in the parallel and perpendicular directions to the stretching direction, the corresponding 1D cross-sections were obtained by using the angle-selective integration

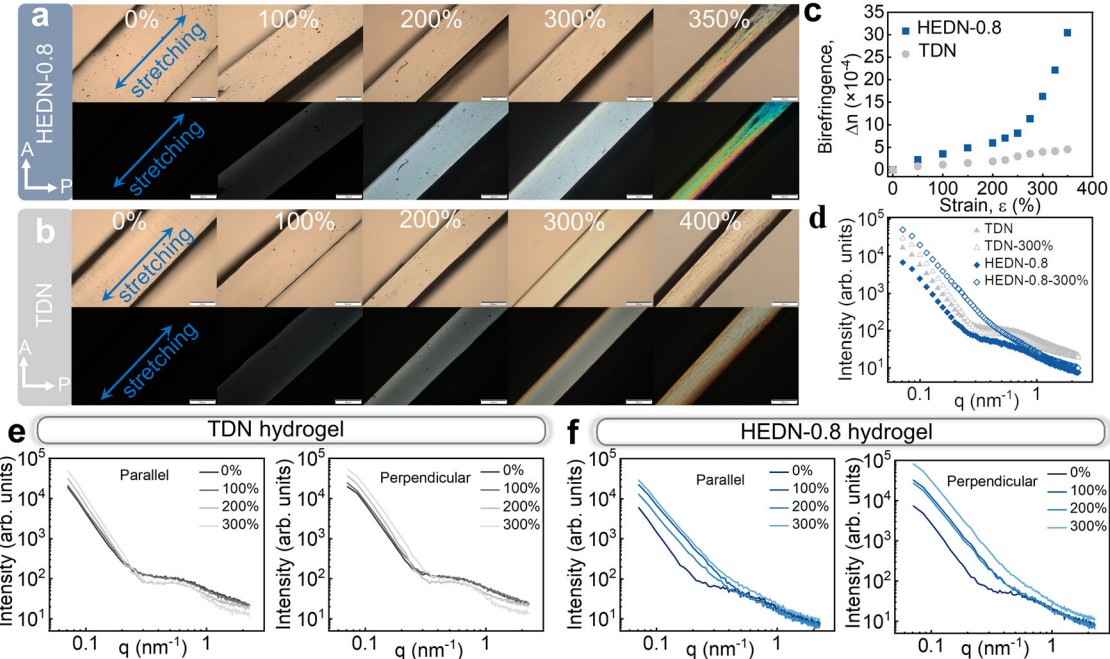

**Fig. 4 | Self-reinforced mechanism and uniformity characterization of HEDN-0.8 hydrogel.** Polarized optical images of HEDN-0.8 (**a**) and TDN (**b**) hydrogel with increasing strains. Photos with a yellow background are regular light microscope images, those with a black background were taken in polarized mode. The stretching direction of the hydrogel samples forms an angle of 45° to the polarizer and analyzer. The direction of A: analyzer, P: polarizer. Scale bars, 500 μm. **c** Normalized birefringence of TDN and HEDN-0.8 hydrogels with increasing strains. **d** SAXS scattering intensity (I) vs scattering vector (q) profiles of TDN and HEDN-0.8 hydrogels with different strains. SAXS profiles of TDN (**e**) and HEDN-0.8 (**f**) hydrogels with different strains in the perpendicular and parallel directions.

method. The shoulder on the scattering profile of the TDN hydrogel shifts slightly with increasing strain. The shoulder on the parallel profile shifts to the low-q region, while the shoulder on the perpendicular profile shifts to the high-q region. Due to the elongation of the hydrogel samples in the parallel to stretching directions, resulting in an increase in the spacing of the microdomains. In the perpendicular direction, the hydrogels become thinner, causing the microdomains to becomes closer to each other. In contrast, the scattering of HEDN-0.8 hydrogel becomes weaker with increasing strain in both parallel and perpendicular directions, indicating that stretching leads to the gradual disappearance of microdomains due to the sliding effect of polymer chain entanglements. SAXS profiles and patterns also indicate that HEDN hydrogels have high recoverability (Supplementary Fig. 21). The SAXS profiles before and after stretching show almost exactly overlapping curves, indicating that the HEDN hydrogels evolved from a non-uniform initial state to a uniform network structure after stretching, and then returned to a non-uniform structure after unloading. This recovery from a highly ordered stretch structure to a disordered entanglement structure is driven by entropy. Moreover, there is almost no energy dissipation structure in the network. Thus, HEDN hydrogels have high recoverability.

The stretched HEDN-0.8 hydrogel network is highly uniform oriented due to the sliding of entanglement points during stretching. This highly uniform orientation structure endows high water content hydrogels with high strength and high strain-stiffening capability. This entropic elastic nature without energy dissipation also gives these hydrogels high reversibility and fatigue resistance.

**Stress dispersion mechanism at the notch of HEDN hydrogels**
The theory of fracture mechanics states that due to the singularity in stress and strain, the deformation field near the crack tip is usually much greater than that in the bulk[53–55]. Thus, it may be possible to improve the fracture energy of a material by affecting the deformation field at the crack tip. In Fig. 2c, the fracture energies of TDN hydrogel

and HEDN hydrogels are significantly different. The fracture energy of the TDN hydrogel is only 385 J m⁻², while that of the HEDN-0.8 hydrogel reach 8340 J m⁻². This shows that the HEDN hydrogels prevent crack propagation. Comparing Supplementary Movies 3 and 4 also shows that cracks in the notched TDN hydrogel expand under a very small strain, while those in notched HEDN hydrogels are blunted under a small strain, and crack growth only occurs under high strain. Due to the anchoring of polymer chains by the high-density chemical cross-linking sites in the first network of the TDN hydrogel, only the first elongated polymer chain bears the force of stretching, while other polymer chains remain in their original state. When the first polymer chain breaks, stress is transferred to other chains, and this breakage causes a chain reaction in the hydrogel network that cause the crack to rapidly expand (Fig. 5a, c). The polarized light microscope observed that the curled polymer chains were oriented along the direction of the force when they were stressed, and interference colors were produced under crossed polarized light. As shown in Fig. 5b, the interference color increases gradually when the notched TDN hydrogel is stretched from 0% to 60% strain (the crack begins to grow at 60% strain). However, it is mainly concentrated near the notch, which indicates that only the polymer chains around the notch are stretched, while other polymer chains remain unstressed or un-oriented. These are a typical stress concentration phenomenon. On the contrary, the HEDN hydrogels not only appears interference color around the notch and bright interference color at other positions, which indicates the stress is distributed to the bulk. In addition, the notched HEDN hydrogels can be stretched to 260% before notch growth. Supplementary Movies 5 and 6 of the notched hydrogels more directly reflect this stress dispersion phenomenon during in situ stretching, implying that notched HEDN hydrogels can recruit more polymer chains to withstand greater external forces. The mechanism is shown in Figs. 5d and 5f, the densely entangled points begin to slip and distribute the force to other chains, which avoids stress concentration and recruiting more chains bear external force. Therefore, the effective stress dispersion induced by

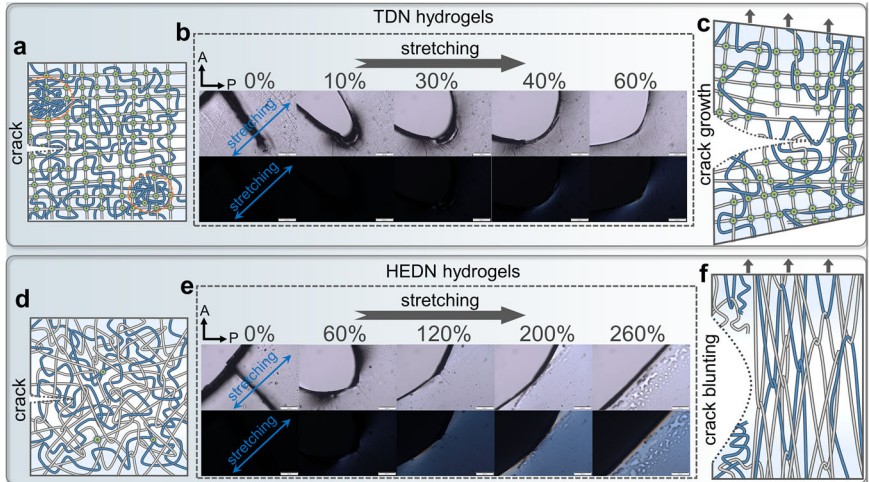

**Fig. 5 | Crack blunting mechanism of HEDN hydrogels.** Illustration of the network structure of the notched TDN (**a**) and notched HEDN (**d**) hydrogels. Polarized optical images of notched TDN (**b**) and notched HEDN (**e**) hydrogel with increasing strains, showing the absence of crack propagation. Photos with a white background are regular light microscope images, and those with a black background were taken in polarized mode. The stretching direction of the hydrogel samples forms a 45° angle with the direction of the polarizer and analyzer. The direction of A: analyzer, P: polarizer. Scale bars, 500 µm. Illustration of network structure for the notched TDN (**c**) and notched HEDN (**f**) hydrogels in the stretched state.

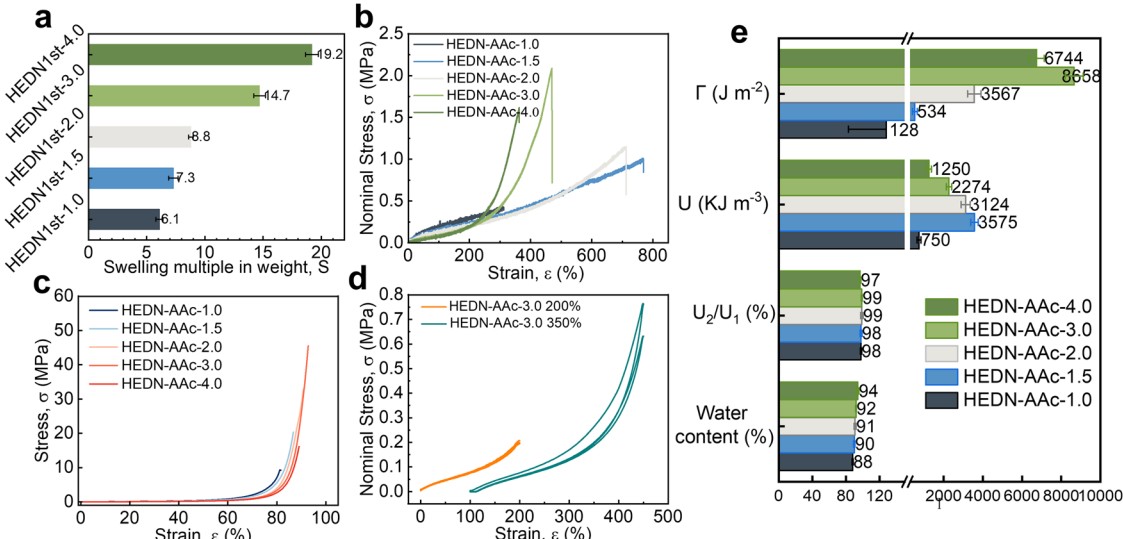

**Fig. 6 | Applicability of highly entangled double network strategy to AAc system. a** Swelling multiple in weight of HEDN1st with different AMPS contents in 4 M AAc solution. **b** Tensile stress-strain curves of HEDN-AAc hydrogels with different AMPS contents. **c** Compressive stress-strain curves of HEDN-AAc hydrogels with different AMPS contents. **d** Twice consecutive stretching curves of HEDN-AAc-3.0 hydrogel in 200% and 350% strain, respectively. **e** Water content, reversibility ($U_2/U_1$), toughness (U) and fracture energy ($\Gamma$) of HEDN-AAc hydrogels with different AMPS contents. Error bars represent mean +/- standard deviation ($n = 5$).

entanglement around the notch gives HEDN hydrogels ultrahigh fracture energy.

## Application of the highly entangled double network strategy

The acrylic-based highly entangled double-network (HEDN-AAc) hydrogels obtained by changing the second network monomer from AAm to AAc retain the above characteristics. Because the ratio of the first network to the second network is one of the core parameters determining the performance of DN hydrogels, the swelling ratio of HEDN1st hydrogels soaked in the second network monomer solution was investigated in detail. As shown in Fig. 6a, the HEDN1st-1.0 network only has a swelling multiple of 6.1 times in 4 M AAc solution, but it can swell 16.5 times in neutral 4 M AAm solution. This is attributed

to the weak electrolyte nature of AAc. The swelling multiple of HEDN1st hydrogels gradually increased upon further increasing the AMPS content, and the swelling multiple of HEDN1st-4.0 hydrogel was as high as 19 due to the strong electrostatic repulsion between the high-density polyelectrolyte chains. HEDN-AAc hydrogels prepared with AAc as the second network monomer still have good mechanical properties. Figure 6b shows that the tensile strength of HEDN-AAc hydrogels first increases and then decreases with swelling multiple, HEDN-AAc-3.0 hydrogel has the highest tensile strength of 2.1 MPa, and the strain-stiffening phenomenon is gradually obvious. The compressive strength of HEDN-AAc hydrogel also showed a similar trend, HEDN-AAc−3.0 hydrogel exhibited a compressive strength of 45 MPa. Figure 6d shows that the HEDN-AAc-3.0 hydrogel

also exhibits good reversibility at 200% strain. It is worth noting that HEDN-AAc-3.0 hydrogel also has high water content and fracture energy, which are 92% and 8658 J m$^{-2}$, respectively (Fig. 6e). In summary, the highly entangled double network hydrogels obtained by using AAc as the second network also show high water content, high strength, high toughness, high reversibility, and strain-stiffening, which expands the application range of the highly entangled double network strategy.

## Discussion

In summary, we developed highly entangled double network (HEDN) hydrogels with high mechanical performance, in which physical entanglements acted as the primary effective crosslinking in the first network. The sliding entanglement points endow HEDN hydrogels with a nearly uniform orientation network structure when stretched, which gives HEDN hydrogels typical strain-stiffening characteristic and high tensile strength. Moreover, change in the hydrogel network based on entanglements sliding distributes the stress concentrated at the notch to the bulk, which gives the hydrogel high fracture energy. There is no energy dissipation in the hydrogels when it is stretched, and the energy is stored in the oriented polymer chains as an entropy loss. HEDN hydrogels with high reversibility and fatigue resistance due to the nature of this entropic elasticity. The highly entangled double network scheme provides a promising strategy for developing hydrogel materials with a high-water content, high toughness, high reversibility, fatigue resistance, and strain-stiffening characteristics. More significantly, this is a successful application of entanglement network in high-performance hydrogels, which leads to further understanding of the role of entanglements.

## Methods

### Materials

Acrylamide (AAm, ≥98%), 2-acrylamido-2-methyl-1-propanesulfonic acid (AMPS, 99%), N,N′-methylenebisacrylamide (MBAA, ≥99.5), 2-Hydroxy-4′-(2-hydroxyethoxy)-2-methylpropiophenone (Irgacure 2959, 98%), were received from Sigma-Aldrich Chemistry Co. Acrylic acid (AAc, 99%), was purchased from Aladdin Chemistry Co., Ltd. All chemical reagents are used as supplied without any purification. Millipore deionized water (18.2 MΩ cm$^{-1}$) was used as a solvent.

### Preparation of HEDN 1st hydrogels

The detailed preparation formula of the first network of highly entangled double network (HEDN) hydrogels is shown in Supplementary Table 1. In short, appropriate amounts of AAm and AMPS were dissolved in a very small amount of deionized water, then the crosslinker MBAA and initiator I2959 were added and mixed well to obtain the prepolymer solution of the highly entangled double network first network (HEDN1st) hydrogels solution, followed by bubbling nitrogen gas for 15 min to remove dissolved oxygen in the solution. The prepolymer solution was injected into a glass plate mold containing a 0.5 mm thick silica gasket, and the front and back were irradiated under a 365 nm ultraviolet lamp for 2 h to complete the polymerization reaction. The HEDN1st hydrogels used in the test need to be swelled in deionized water for 24 h to reach equilibrium. Moreover, four HEDN1st-0.8 hydrogels with different degrees of entanglement were prepared as the first network, and the detailed formulations are shown in Supplementary Table 2.

### Preparation of HEDN (HEDN-AAc) hydrogels

Soak the prepared HEDN1st hydrogels in the second network prepolymer solution for 24 h to absorb enough monomers. The composition of the second network prepolymer solution used to prepare HEDN hydrogels (HEDN-AAc hydrogels) was 4 M AAm (AAc), 0.05 mol% crosslinker and 0.01 mol% initiator. The HEDN1st hydrogels having absorbed enough second network monomers were

irradiated under a 365 nm ultraviolet lamp for 4 h to form the second network in situ. Finally, the resulting hydrogels were placed in pure water to obtain balanced swollen HEDN hydrogels (HEDN-AAc hydrogels).

### Preparation of TDN hydrogel

The preparation process of traditional double-network (TDN) hydrogel is described in previous reports[2]. Briefly, an appropriate amount of AMPS was dissolved in deionized water to form a 1 M monomer solution, and then 4 mol% of crosslinker MBAA and 0.1 mol% of initiator I2959 were added. The above solution was mixed thoroughly and deoxidized by N$_2$, then poured into the mold, and irradiated under a 365 nm ultraviolet lamp for 4 h to obtain the first network of the traditional double network hydrogel. Subsequently, the network was immersed in a second network monomer solution consisting of 4 M acrylamide, 0.05 mol% cross-linking agent, and 0.01 mol% initiator for 24 h, followed by polymerization of the second network under ultraviolet light for 4 h, and the obtained pre-prepared gel was soaked in deionized water for 24 h to obtain a traditional double network hydrogel (TDN hydrogel) with swelling equilibrium.

### Measurement of swelling multiple in weight

The first network hydrogels with a mass of $m_1$ were soaked in the second network monomer solution for 24 h, the mass of the swollen hydrogels was $m_2$, and the swelling multiple in weight $S$ was calculated by the following formula:

$$S = m2/m1 \qquad (1)$$

### Measurement of water content

The swelling equilibrium hydrogel samples whose mass was $m_a$ were freeze-dried and the mass of the dry gel after removing all water was recorded as $m_b$, and the water content was calculated by the following formula:

$$WC = \frac{m_a - m_b}{m_a} \times 100\% \qquad (2)$$

### Preparation of gels with specific water content

A quantitative amount of hydrogel with known water content or completely dehydrated gel is placed in a sealed box. After adding an appropriate amount of water, it is placed in a constant-temperature oscillator at 37 °C and incubated for 48 h to obtain a gel with a determined water content.

### ATR-FTIR characterizations

The binding information of characteristic groups was studied using an infrared spectrometer (PerkinElmer Spectrum 100) with attenuated total reflection (ATR) accessory. Before testing, the hydrogels should be freeze-dried.

### Transmittance characterization

The visible light (400–800 nm) transmittance of the hydrogel samples was measured using a UV-Vis-NIR spectrophotometer (PerkinElmer LAMBDA 950).

### Mechanical Test

Uniaxial tensile and compressive mechanical testing of hydrogel samples were done using a universal tensile machine (MTS C43.104) equipped with 500 N and 10000 N load cells. The samples for uniaxial stretching were cut into a dumbbell-shaped spline with a length of 30 mm and a width of 5 mm, and the thickness was accurately measured with a vernier caliper. All uniaxial tensile tests were done at room

temperature at a tensile speed of 50 mm min$^{-1}$. The nominal stress is defined as the applied load force divided by the initial area of the sample cross-section. The strain is calculated by this formula: $\varepsilon = \frac{L_1 - L_0}{L_0} \times 100\%$ (3), where $L_1$ is the distance of the clamp when the sample breaks, $L_0$ is the initial distance of the clamp, at least three parallel experiments for each group of samples. The nominal stress and strain of the samples can be obtained directly from the stress-strain curve, the modulus (stiffness) is the slope of the stress-strain curve when the strain is 5%, and the fracture work (toughness) is the area enclosed by it. The reversibility of the hydrogels is defined as the ratio of the area covered by the second cycle loading curve to the area covered by the first cycle loading curve, $U_2/U_1$, hysteresis rate is $1 - U_2/U_1$. The differential modulus versus strain curve ($\partial\sigma/\partial\lambda$-$\varepsilon$) of the sample is obtained by fitting the stress-strain curves with multiple differential moduli under different strains, and the strain-stiffening capacity is the ratio of the maximum differential modulus to the minimum differential modulus. The samples need to be coated with silicone oil to keep moisture during cyclic stretching, and the stretching rate was 500 mm min$^{-1}$ without any termination. The hydrogels used for uniaxial compression were cylindrical samples with a diameter of 10 mm and a thickness of 7–10 mm, and the compression rate is 5 mm/min.

### Pure shear test

Pure shear testing was used to determine the fracture energy of hydrogel samples. The samples were made into a 30 mm × 20 mm rectangular spline, and a 10 mm notch was introduced on one side in the middle of the samples. The fracture energy is calculated by this formula: $\Gamma = HU(\lambda_c)$ (4), $H$ is the initial distance of the clamp, $\lambda_c$ is the critical strain when the crack begins to propagate in the notched sample, and $U(\lambda_c)$ is the area of the stress-strain curve of the unnotched sample under the $\lambda_c$ strain.

### Scanning Electron Microscope characterization

The unstretched and stretched 300% hydrogel samples were frozen in liquid nitrogen and quickly put into a freeze dryer for further freeze-drying to obtain xerogels. After a layer of platinum was sputtered on the surface of the sample, their morphology was observed with a scanning electron microscope (SEM). The hole size in the images was counted using the software Nano Measure.

### Birefringence test

A polarizing microscope (Olympus BX53-P) with a 4× objective lens was used to take pictures of samples with or without notch under different tensile strains. The birefringence index ($\Delta n$) was measured at the center of the hydrogel using the Berek compensator and calculated with this formula: $\Delta n = $ retardation (μm)/sample thickness (μm) (5). In situ stretching movies of notched and unnotched samples were recorded with the help of a self-made manual tensioner.

### Small-angle X-ray scattering

Rectangular hydrogel specimens with dimensions of 30 mm × 8 mm were used for SAXS (NanoSTAR, Bruker AXS, Germany) testing. The samples were perpendicular to the beam, and the irradiation time was 10 min. The stretching direction of the samples is also perpendicular to the beam, the stretching rate was 50 mm/min, and the scattering vector $q$ ranged from 0.07 to 2.3 nm$^{-1}$. The distance between microdomains was calculated by the formula $d = 2\pi/q$ (6). The SAXS patterns were radially averaged to obtain the intensity profiles. 1D profiles perpendicular and parallel to the stretching direction were also acquired.

### Atomic force microscope characterization

The surface morphology of the hydrogel samples was observed and the roughness Rq was calculated using a biological atomic force microscope (FastScan Bio, Bruker, USA). The samples needed to be immersed in the water-liquid pool during the measurement.

## Data availability

The data are available within the article and Supplementary Files. The other relevant source data are available at https://doi.org/10.6084/m9.figshare.24523249, or available from the corresponding authors on request.

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

## Acknowledgements

This study was financially supported by National Nature Science Foundation of China (No. 22075177, X.W.). We thank engineer Fang Zhou from Evident (China, Shanghai) Co., Ltd. for the help with polarizing measurements.

## Author contributions

R.Z. designed the study, synthesized hydrogels, conducted experiments and prepared the manuscript. D.Z. conducted the analysis and manuscript writing. X.W. and Z.Z. contributed to the manuscript revision and data analysis.

## Competing interests

The authors declare no competing interests.
