## [Peer Review File · Nature Communications]

Tough double network hydrogels with rapid self-reinforcement and low hysteresis based on highly entangled networksREVIEWER COMMENTS

Reviewer #1 (Remarks to the Author):

In this manuscript, the authors propose a straightforward approach for fabricating highly entangled double network (HEDN) hydrogels with a more homogeneous structure. The fabrication of HEDN hydrogels is similar to that of traditional double-network hydrogels. Unlike traditional double-network hydrogels, the first network from the random copolymerization of neutral monomer (AAM) and ionic monomer (AMPS) in HEDN forms the highly topological entanglements, regarding as effective crosslinking points. The author stated that the presence of free-sliding entanglements imparts the hydrogel network with exceptional mechanical strength and low hysteresis. However, its novelty should be reevaluated. In conclusion, this version of the manuscript is not suitable for publication in this journal. Detailed comments are provided below:

(1) Firstly, my understanding is that HEDN hydrogels primarily fall under the category of interpenetrating networks rather than the traditional double-network concept. Secondly, the fabrication method for HEDN hydrogels has been reported in the previous work (Adv. Funct. Mater. 2012, 22, 4426–4432), which dilutes the novelty. The distinction between this work and previously reported work should be presented in the introduction part.

(2) The authors suggest that the low hysteresis and high strength of HEDN hydrogels originate from highly topological entanglements within the first network. However, there is no direct evidence to support that first network is highly topological entanglements.

(3) The manuscript describes the structure of hydrogels using SEM analysis, conducted by freezing the structure in liquid nitrogen. However, it is unclear whether the observed structure is representative of the original structure of the hydrogel or related to the dimensions associated with ice crystal formation during this sample preparation process.

(4) The authors claimed that HEDN hydrogels exhibit a more uniform network structure, which contributes to their superior mechanical properties compared to TDN hydrogels. However, the SAXS profile depicted in Figure 4f displays an apparent shoulder peak in the undeformed state of HEDN hydrogel, indicating the presence of structural inhomogeneities within the HEDN hydrogel.

(5) In the legend of Figure.4 and title of this manuscript, it is not specified which experimental data represents the “Self-reinforced behavior” of HEDN hydrogels.

Reviewer #2 (Remarks to the Author):

In this manuscript, Zhu et al. reported the synthesis and characterization of double network hydrogels (DN gels) with high toughness while low hysteresis based on high entanglement. The HEDN gels showed high strength of 3 MPa, high fracture energy of 8340 J/m², high strain-stiffening capability of 47.5 and low hysteresis. Although there are some interesting results, I don't think it can be accepted in current state. The main mechanism based on homogeneous networks is misled in concept. Also, I have some

questions about this work:

1. The first network of HEDN gels is a PAAm gel with dense physical entanglement, which had been reported by Suo's group (Science, 2021, 374, 212-216). Using this gel or AMPS modified gel as first network, the HEDN gels were prepared by the further introduction of another PAAm network as second network. Because the first PAAm is already highly entangled, which will bear most of the force during deformation. What's the role of the second PAAm network? If it does work, the monomer concentration and chemical cross-linkers content will also affect the entanglement, however, these data cannot be found in the manuscript?
2. In Suo's work, they suggested the entanglements constrain swell of the PAAm gel. Also, it's found in Figure 2a that the HEDN-0.2 exhibited the similar mechanical properties to HEDN1st-0.2 in Figure 1d, and the swelling of HEDN1st-0.2 in Figure 1c is the smallest. To improve the swelling of the HEDN1st, the AMPS comonomer is introduced in the HEDN1st. How does the swelling influence the entanglement? Can the entanglement be decreased by the swelling? In Figure 1d, the mechanical properties of HEDN1st should be compared with the same level of water content.
3. In Figure 2a and the main text, TDN gel was used as the control to compare with HEDN gel, however, the TDN gel prepared in this work is much weaker (< 1 MPa and $< 450\%$) and worse toughness (< 400 J/m²) than those reported by Prof. Gong's group, why? I don't think it's a good control.
4. Since both HEDN1st and HEDN are highly entangled. The comparison between HEDN1st and HEDN with the same polymer volume fraction should be provided. At the same polymer volume fraction, HEDN1st may show similar mechanical properties to HEDN. If so, the second PAAm network seems useless.
5. In Figure S7 and S8, it illustrated the gels show strain stiffening, what's the mechanism of the strain stiffening? Why there is an optimal value to HEDN-0.8? Higher swelling of HEDN1st should be adsorbed more second network monomer and should be shown higher entanglement, why the strain-stiffening capability is decreased after HED1st-0.8?
6. The authors have the wrong understanding of "homogeneous network". The SEM and SAXS were used to interpret the homogeneous network. Up to date, only the tetra-PEG hydrogels are considered as near-homogeneous network. Also, what's the meaning of "homogenous orientation"? The concept is unclear.
7. In Figure 4d, it's clearly found there is a peak of 0.6 nm^{-1} (correlation length of $\sim 10.46 \text{ nm}$) in HEDN-0.8, indicating there is chain aggregation in the HEDN. The aggregation indicates the HEDN isn't a homogenous network. And, why is there chain aggregation? The scale of chain entanglement cannot explain such large aggregation.
8. Also, in Figure 4d and 4f, the intensity after stretching increased, why?
9. The HEDN gel demonstrated good recovery with low hysteresis. Therefore, I think the peak should be found again after unloading.
10. Compared to HEDN1st with a fatigue threshold of ~ 200 J/m², can this DN structure improve the fatigue threshold?

Point-by-point response to the reviewers' comments

Reviewer #1 (Remarks to the Author):

In this manuscript, the authors propose a straightforward approach for fabricating highly entangled double network (HEDN) hydrogels with a more homogeneous structure. The fabrication of HEDN hydrogels is similar to that of traditional double-network hydrogels. Unlike traditional double-network hydrogels, the first network from the random copolymerization of neutral monomer (AAm) and ionic monomer (AMPS) in HEDN forms the highly topological entanglements, regarding as effective crosslinking points. The author stated that the presence of free-sliding entanglements imparts the hydrogel network with exceptional mechanical strength and low hysteresis. However, its novelty should be reevaluated. In conclusion, this version of the manuscript is not suitable for publication in this journal.

Answer: Dear reviewer, we sincerely thank you for your time and effort in reviewing our paper. We have revised the manuscript and added new data after carefully considering your comments and suggestions. All modifications are shown in red in the revised manuscript.

Detailed comments are provided below:

(1) Firstly, my understanding is that HEDN hydrogels primarily fall under the category of interpenetrating networks rather than the traditional double-network concept.

Answer: Thank you for your comments. In the previous manuscript we have a less detailed description. As one of the interpenetrating network hydrogels, the traditional double-network hydrogels are composed of highly chemically crosslinked polyelectrolyte network and loosely crosslinked neutral polyacrylamide network (Chem. Rev. 2021, 121, 4309-4372). In this work, the first network of HEDN

hydrogels is also highly cross-linked, but this cross-linking is mainly served by physical entanglements, and the second network is also a loosely crosslinked neutral polyacrylamide network. Traditional double-network hydrogels are toughened by covalent bond breaking of the first network, HEDN hydrogels are by stretching-induced highly ordered polymer chain orientation. Although the toughening mechanisms of the two hydrogels are completely different, HEDN hydrogels can be regarded as new double-network hydrogels considering the similarity of their network structures. We've added a new description in revised manuscript. (See the second paragraph on Page 4 in the revised manuscript)

Revisions:

Highly entangled double-network (HEDN) hydrogels can be regarded as new DN hydrogels considering that its network structure is similarity to traditional double-network (TDN) hydrogels. ...

Secondly, the fabrication method for HEDN hydrogels has been reported in the previous work (Adv. Funct.Mater.2012, 22, 4426–4432), which dilutes the novelty. The distinction between this work and previously reported work should be presented in the introduction part.

Answer: Thank you for your careful review of our manuscript. The similarity of the preparation methods of St-DN gel (Adv. Funct. Mater. 2012, 22, 4426–4432) and HEDN gel is using high osmotic pressure of polyelectrolyte. This makes the neutral first network swell strongly to absorb the second network monomer and then obtain the DN hydrogels. However, there are significant differences between St-DN gel and HEDN gel in network structure, mechanical properties and research purposes.

Firstly, the first network of HEDN gel is prepared in a solution of high concentrated monomers ($\sim 28 \text{ mol L}^{-1}$) and few chemical crosslinkers (0.001 mol%) and initiators (0.0004 mol%). The monomer concentration of the first network of St-DN gel is 1

mol L⁻¹, and the dosage of chemical crosslinker and initiator are 4 mol% and 0.1 mol%, respectively. Due to the huge difference in these parameters, the main crosslinking sites of first networks in HEDN and St-DN gels are physical entanglement and chemical crosslinking, respectively. This is the first reported method for the synthesis of tough DN gels with dense physical entanglement acting as the main crosslinking.

Secondly, the mechanical properties of HEDN hydrogels and St-DN hydrogels are significantly different due to the obvious difference in network structure. HEDN hydrogels exhibit typical strain-stiffening behavior and very low hysteresis.

Last, the fabrication method for St-DN gels is developed in order to expand the variety of double-network hydrogel. The development of HEDN is to overcome the contradiction between high toughness and low hysteresis of hydrogels.

In conclusion, although HEDN hydrogels and St-DN hydrogels have some similarities in fabrication methods, their distinctions are greater. Moreover, exploiting the high osmotic pressure of polyelectrolytes seems to be a universal strategy. We consider it more appropriate to refer to previously work in the Result part. The preparation principle of St-DN hydrogels should be referred to. (See the second paragraph on Page 4 in the revised manuscript)

Revisions:

... AMPS is introduced into the polymer backbone in order to **increase the osmotic pressure of the polymer network** so that the polymer networks can absorb more second network monomers⁴⁴.

(2) The authors suggest that the low hysteresis and high strength of HEDN hydrogels originate from highly topological entanglements within the first network. However, there is no direct evidence to support that first network is highly topological entanglements.

Answer: Thanks for your precious comments. Yes, we should make this important point clearer. As Suo et al. (Science, 2021, 374, 212-216) and Li et al. (Nature, 618, 740–747 (2023)) commented, entanglements can significantly affect the stiffness of hydrogels and the polymer content at swelling equilibrium. Therefore, the determination of the stiffness of those hydrogels at the same polymer content and the water content of the fully swollen hydrogel can reflect the entanglement degree of the polymer network. The new results have now been included in the revised manuscript and Supplementary Information. (See the second paragraph on Page 4 in the revised manuscript)

Revisions:

The entanglement degree of HEDN1st hydrogels is controlled by the concentration of prepolymer solutions. Entanglements will significantly affect the equilibrium water content of hydrogels. For example, the water content of HEDN1st hydrogels increases significantly with the increase of W, eventually reaching equilibrium at ~99.7%, while the water content is 95.4% when W is 2 (Fig. S1). The swelling rate also shows a similar phenomenon (Fig. S2). Many entanglements effectively inhibit the swelling of networks. The entanglements also increase the stiffness of hydrogels^{9,20}. The HEDN1st hydrogels obtained at five different concentrations were processed into samples with a water content of 89% to eliminate the influence of polymer contents on stiffness. As shown in Fig. S3 (pink area), the stiffness of HEDN1st increases significantly as the W value decreased, indicating a significant increase in physical entanglements. The tensile strength also increases with the increase of entanglement, but the elongation at break is opposite (Fig. S4). Therefore, HEDN1st prepared at the highest monomer concentration (W2) are regarded as highly entangled hydrogels....

Fig. S1 Water content of fully swollen HEDN1st-0.8-W hydrogels with different value of W. W is the molar ratio of water to total monomer.

Fig. S3 Effects of prepolymer concentration, crosslinker dosage and AMPS dosage on stiffness of HEDN1st and HEDN hydrogel, respectively.

Fig. S4 Tensile stress-strain curves of HEDN1st-0.8 hydrogels with different degree of entanglements. Water content: 89%.

(3) The manuscript describes the structure of hydrogels using SEM analysis, conducted by freezing the structure in liquid nitrogen. However, it is unclear whether the observed structure is representative of the original structure of the hydrogel or related to the dimensions associated with ice crystal formation during this sample preparation process.

Answer: Thanks for your comments. As you commented, due to the formation of ice crystals, the morphology of the freeze-dried hydrogel is not exactly the same as that of the wet hydrogel. The holes shown in the SEM images are caused by the formation and sublimation of ice crystals. However, this can still reflect the polymer network distribution of the wet hydrogel. The polymer network distribution in the wet hydrogel is fixed the moment it is immersed in liquid nitrogen. At this time, ice crystals form rapidly, and the polymer chains around the ice crystals are squeezed together to form a hole structure. However, where the polymer network is dense, pores are formed that are small in size but densely distributed (marked by the blue circle in Fig. 3a), and in places where the network is sparse, pores are sparsely distributed but larger in size (marked by the orange circle in Fig. 3a). If the wet hydrogel has a more uniform polymer network, the size and distribution of its pores will be more consistent and uniform after freeze-drying. Therefore, the distribution of the network in the wet hydrogel can still be analyzed by SEM.

The SAXS and AFM analyses of wet hydrogels are consistent with the SEM results above. The non-uniform distribution of network in wet hydrogels will inevitably lead to the difference of electron cloud density. As shown in Figure 4d, the shoulder peak can be clearly observed in the SAXS profile of the TDN hydrogel, which is precisely caused by the uneven network. The shoulder peak was also observed in HEDN hydrogels, but is significantly weakened compared to TDN, indicating that the network became more uniform. This is basically consistent with the phenomena reflected by SEM (Fig. 3). The surface roughness measurement of the hydrogel immersed in the liquid pool also showed that the HEDN hydrogel had a smaller roughness (27.4 nm), and the surface roughness of TDN hydrogel is as high as 133.2

nm (Fig. S12). AFM analysis is also consistent with the phenomena shown by SEM.

Fig. 3 Structural changes of hydrogels before and after stretching. a Microstructure of unstretched freeze-dried TDN hydrogel. Larger or smaller void structures can be observed at the orange or blue circle marks. b Microstructure of freeze-dried TDN hydrogel stretched to 300% of its original length. c-d Diagram of structural changes in the TDN hydrogel before (c) and after (d) stretching. e Microstructure of unstretched freeze-dried HEDN-0.8 hydrogel. f Microstructure of freeze-dried HEDN-0.8 hydrogel stretched to 300% of its original length. g-h Diagram of structural changes of the HEDN-0.8 hydrogel before (g) and after (h) stretching. Scale bars in SEM images: 30 μm .

(4) The authors claimed that HEDN hydrogels exhibit a more uniform network structure, which contributes to their superior mechanical properties compared to TDN hydrogels. However, the SAXS profile depicted in Figure 4f displays an apparent shoulder peak in the undeformed state of HEDN hydrogel, indicating the presence of structural inhomogeneities within the HEDN hydrogel.

Answer: Thanks for your comments. As you mentioned, the SAXS profile of HEDN

hydrogel shows obvious shoulder peak, indicating that there is still an uneven structure. We recognize that. However, it can be clearly seen from Figure 4d the shoulder peak of HEDN hydrogel is significantly weakened compared with undeformed TDN, indicating a more uniform network structure within the HEDN hydrogel.

Moreover, after much reflection, we now believe that stretching is more important for the evenness of polymer networks, which can be called stretching-induced highly uniform orientation. From Figure 4f, it is found that the shoulder peak rapidly disappears with the increase of strain, which indicates that a highly uniform oriented structure is formed. For TDN hydrogels, the shoulder peak does not change with the change of strain, indicating that the TDN hydrogel still has an uneven structure after stretching. Thus, stretching only aligns the polymer network in the TDN hydrogel, but does not improve the uneven distribution of its network, which is determined by the fixed chemical crosslinking points. Stretching can not only orient the network of HEDN hydrogels, but also adjust the network structure to be highly uniform by the sliding entanglements.

We realize that it is unreasonable to attribute excellent performance of HEDN hydrogels simply to its uniform network, so the manuscript has been comprehensively revised.

Revisions:

Title: Tough double network hydrogels with rapid self-reinforcement and low hysteresis based on **highly entangled** networks

Abstract: ... **high mechanical performance** highly entangled double network (HEDN) hydrogels **without energy dissipation structure** were fabricated ...

Introduction: ... **The free-sliding entanglements cause HEDN hydrogels to form a highly uniform orientation structure under tensile strain, which is summarized as stretching-induced highly uniform orientation behavior.** ...

Results: ... SAXS (Fig. 4d) also shows that both the undeformed TDN and HEDN hydrogels have obvious shoulder peaks, indicating that both have heterogeneous structures, which are caused by ununiform chemical crosslinking and ununiform polymer entanglement, respectively. ...

... This indicates that although there is a certain non-uniform entanglement network in the undeformed HEDN hydrogels, the sliding entanglements make the stretched polymer network achieve uniform orientation. ...

Discussion: ... a nearly perfect uniform orientation network structure when stretched ...

(5) In the legend of Figure.4 and title of this manuscript, it is not specified which experimental data represents the “Self-reinforced behavior” of HEDN hydrogels.

Answer: Thanks for your comments. We have a less detailed description in the previous manuscript. In this work, self-reinforced behavior of HEDN hydrogels refers to a very typical strain-stiffening phenomenon. Figure 2f, S11, and S12 all demonstrate this strain-stiffening phenomenon and are described in the manuscript. “Further analyzing and differentiating the tensile curves of TDN and HEDN hydrogels (Fig. S11), the modulus of TDN hydrogel gradually decreases to a constant value with the increase of strain, while the modulus of HEDN hydrogels increases rapidly from a constant value to a larger value, showing a typical strain-stiffening behavior. And strain-stiffening capacity of HEDN-0.8 hydrogel can reach up to 47.5 times (Fig. S12), which is higher than most hydrogels with strain-stiffening reported (Fig. 2f).” In order to describe this self-reinforced behavior based on strain-stiffening phenomenon in more detail, we have made a new description in the main text. (See the Page 11 in the revised manuscript)

Revisions:

... As shown in Figure S11b, the modulus of HEDN hydrogels continues to increase with strain. The strain-stiffening capacity of HEDN-0.8 and HEDN-1.0 exceeds 40 (Fig. S12), showing strong self-reinforced behavior, which is conducive to avoiding material damage under large strains.

Reviewer #2 (Remarks to the Author):

In this manuscript, Zhu et al. reported the synthesis and characterization of double network hydrogels (DN gels) with high toughness while low hysteresis based on high entanglement. The HEDN gels showed high strength of 3 MPa, high fracture energy of 8340 J/m², high strain-stiffening capability of 47.5 and low hysteresis. Although there are some interesting results, I don't think it can be accepted in current state. The main mechanism based on homogeneous networks is misled in concept. Also, I have some questions about this work:

Answer: Dear reviewer, we sincerely thank you for your time and effort in reviewing our paper. We have carefully corrected misleading descriptions about homogeneous networks and added necessary data. All modifications are shown in red in the revised manuscript.

As you mentioned, the homogeneous network should be similar to the tetra-arm PEG hydrogel network, where the molecular chain length between any two crosslinking points is almost the same, and no defect¹. Recent studies have shown that hydrogel networks are highly spatially inhomogeneous, especially those obtained by radical polymerization^{2,3}. Spatial inhomogeneity refers to the coexistence of densely and loosely crosslinked domains in the network, as shown in Fig. 3a. In this work, the coexistence of two kinds of microdomains is called a non-uniform network, and the opposite is a uniform network. The two kinds of microdomains will have different deformation when stretched, so the network orientation is not uniform. The undeformed HEDN hydrogels also show some non-uniform network due to non-uniform entanglements. However, due to the sliding entanglement, the network can form a highly uniform oriented structure under tensile. Therefore, we now suggest that the formation of highly uniform network structure induced by stretch is the main mechanism for the toughening of HEDN hydrogels.

1. The first network of HEDN gels is a PAAm gel with dense physical entanglement,

which had been reported by Suo's group (Science, 2021, 374, 212-216). Using this gel or AMPS modified gel as first network, the HEDN gels were prepared by the further introduction of another PAAm network as second network. Because the first PAAm is already highly entangled, which will bear most of the force during deformation. What's the role of the second PAAm network? If it does work, the monomer concentration and chemical cross-linkers content will also affect the entanglement, however, these data cannot be found in the manuscript?

Answer: Thanks for your precious comments. Yes, we should make this important point clearer. We think there are two roles of the second PAAm network. First, it provides higher entanglement, and second, there is synergy between the two networks. As Suo et al. (Science, 2021, 374, 212-216) and Li et al. (Nature, 618, 740–747 (2023)) commented, entanglements can significantly affect the stiffness of hydrogels. Therefore, stiffness was used in this work to characterize the entanglement degree of hydrogels network, and other parameters of hydrogels should be consistent. As shown in Figure S3 pink area, HEDN hydrogels have higher stiffness than HEDN1st hydrogels at the same water content, indicating that the second PAAm network further increases entanglements. HEDN hydrogels exhibit unusually high strength compared to HEDN1st hydrogels (Fig. S16a). To eliminate the effects of entanglements, HEDN-0.8-W4 with the same entanglement degree as HEDN1st-0.8 is used for comparison. Fig. S16b shows that the mechanical strength of HEDN hydrogels with double networks is still much higher than that of HEDN1st with single networks. Therefore, we suggest that two independent networks produce significant synergies. The concentration of monomer and the content of chemical crosslinker do affect the entanglement and mechanical properties of hydrogels. We've added that to the revised manuscript. (See the second paragraph on Page 8 in the revised manuscript)

Revisions:

Increasing the concentration of the second monomer solution will significantly enhance the total entanglement of HEDN hydrogel (purple area in Fig. S3), thus

affecting its mechanical properties. As shown in Fig. S15, the tensile strength of HEDN hydrogel first increases and then decreases with the increase of entanglement, which means that the more entanglement is not the better. Within a reasonable range, the increase of chemical crosslinking agents does not affect the stiffness of the hydrogels (green area in Figure S3), which is attributed to the fact that the number of entanglement points is much higher than that of chemical crosslinking, which plays a dominant role in the effect of stiffness. This is consistent with what Suo et al. reported²⁰. However, the mechanical properties of HEDN hydrogels first increased and then decreased with the increase of chemical crosslinkers (Fig. S14), indicating that the appropriate dosage of chemical crosslinkers is also an important factor in achieving high performance of HEDN hydrogels. Finally, we should explain the necessity of double networks. It can be seen from Fig. S16a that the tensile strength of HEDN-0.8 is much higher than that of HEDN1st-0.8 under the same water content. However, due to the introduction of a second network, the entanglement degree of HEDN-0.8 is higher than that of HEDN1st-0.8. In order to exclude this effect, HEDN-0.8-W4 with the same entanglement degree as HEDN1st-0.8 is used for comparison. Fig. S16b shows that the mechanical strength of HEDN hydrogels with double networks is still much higher than that of HEDN1st with single networks.

Fig. S16 a Tensile stress–strain curves of HEDN1st-0.8 hydrogel and HEDN-0.8 hydrogels with the same level of water content. **b** Tensile stress–strain curves of HEDN1st-0.8 hydrogel and HEDN-0.8-W4 hydrogels with the same level of water

content and stiffness.

Fig. S3 Effects of prepolymer concentration, crosslinker dosage and AMPS dosage on stiffness of HEDN1st and HEDN hydrogel, respectively.

Fig. S14 Tensile stress-strain curves of HEDN-0.8 hydrogels with various contents of 2nd network MBAA. Water content: 89%.

Fig. S15 Tensile stress-strain curves of HEDN-0.8 hydrogels with various concentrations of 2nd network monomer. Water content: 84%.

2. In Suo's work, they suggested the entanglements constrain swell of the PAAm gel. Also, it's found in Figure 2a that the HEDN-0.2 exhibited the similar mechanical properties to HEDN1st-0.2 in Figure 1d, and the swelling of HEDN1st-0.2 in Figure 1c is the smallest. To improve the swelling of the HEDN1st, the AMPS comonomer is introduced in the HEDN1st. How does the swelling influence the entanglement? Can the entanglement be decreased by the swelling? In Figure 1d, the mechanical properties of HEDN1st should be compared with the same level of water content.

Answer: Thanks for your comments. Swelling does not affect the total entanglement, but the swelling hydrogels have less entanglement per unit volume. When the prepolymer concentration is the same, the entanglement degree of the resulted hydrogel is almost constant, which can be demonstrated by the blue area in Figure S3. However, due to the introduction of more AMPS, the electrostatic repulsion is enhanced, so the molecular chain will become more stretched, that is, the hidden length is sacrificed, and the macro performance is the swelling of the hydrogel. The total degree of entanglement remains the same, but the volume of the hydrogel increases, so the entanglement per unit volume will be less.

The mechanical properties of HEDN1st with the same level of water content have been corrected, as shown below.

Fig. 1 Highly entangled double network design and first network parameter. a Schematic illustration of the preparation method of the HEDN hydrogels. First, the HEDN1st hydrogels are prepared by copolymerization of AAm, AMPS and MBAA using UV initiated free radical polymerization. Then HEDN1st hydrogels are incubated in the second network monomer solution for 24 h. Finally, the second network is generated in situ to obtain HEDN hydrogels. b FTIR spectra of HEDN1st xerogels with different AMPS contents. c Swelling multiple in weight of HEDN1st hydrogels with different AMPS contents in second network monomer solution. Error bars represent SD. d Tensile stress-strain curves of HEDN1st hydrogels, **water content: 89%**.

3. In Figure 2a and the main text, TDN gel was used as the control to compare with HEDN gel, however, the TDN gel prepared in this work is much weaker (< 1 MPa and < 450%) and worse toughness (< 400 J/m²) than those reported by Prof. Gong's group, why? I don't think it's a good control.

Answer: There is no doubt that the work of Prof. Gong's group has led to the development of high toughness hydrogels, and the toughening mechanism based on

double-network hydrogels has been extended to many fields. This work is also inspired by their work, and some unexpected data are obtained.

We summarize some of the work of Prof. Gong's group on the mechanical properties of DN hydrogels, only the traditional double-network hydrogels with PAMPS as the first network and PAAm as the second network. It can be seen from the Table 1 that the fracture stress of DN hydrogel is 0.2-1.4 MPa and the elongation at break is 400-2000%. Second network in these works typically have a monomer concentration of 2 mol/L and a crosslinker dosage of 0.01 mol% or 0.02 mol%. In this work, in order to be consistent with the second network of HEDN, the monomer concentration of the second network is 4 mol/L and the dosage of crosslinker is 0.05 mol%. A higher monomer concentration means more physical entanglement, and an increase in physical entanglement and chemical crosslinking agents will reduce the elongation at break. The fracture strength of TDN hydrogel prepared in this work is 0.8 MPa and the elongation at break is 440%, which is basically consistent with the work reported by Gong's group. The toughness of TDN hydrogel is also consistent with 10^2 - 10^3 J/m² reported by Gong's group^{4,5}. To avoid misunderstandings, we added a note to the legend in Fig. 2.

Finally, compared with chemical crosslinking points, TDN is necessary as a control to reflect the role of physical entanglement in toughening hydrogels, which is conducive to further understanding of entanglement networks.

Table 1. Some parameters of the work reported by Gong's group on DN hydrogels

	Stress (MPa)	Strain (%)	1st monomer concentration (mol/L)	1st crosslinker concentration (mol%)	2nd monomer concentration (mol/L)	2nd crosslinker concentration (mol%)
This work	0.8	440	1	4	4	0.05
J. Am. Chem. Soc. 2023, 145, 7376-7389	1.1	1700	1	3	2	0.01
ACS Macro Lett. 2019, 8, 11, 1407-1412	1.3	1100	1	4	4	0.01
Polymer 55 (2014) 914- 923	0.6	800	1	4	2	0.02
J. Am. Chem. Soc. 2022, 144, 3154-3161	0.2	2000	1	4	2	0.02
Extreme Mechanics Letters 51 (2022) 101588	1.4	2000	1	3	2	0.01
Macromolecules 2010, 43, 22, 9495-9500	0.7	400	1	3	2	0.1
Soft Matter, 2020,16, 5487-5496	1.0	500	1	4	2	0.02
	0.8	800	1	4	2	0.02

Revisions:

... Note: Preparation Parameters of TDN hydrogel: 1st AMPS (1M) MBAA (4 mol%), 2nd AAm (4M) MBAA (0.05 mol%).

4. Since both HEDN1st and HEDN are highly entangled. The comparison between HEDN1st and HEDN with the same polymer volume fraction should be provided. At the same polymer volume fraction, HEDN1st may show similar mechanical properties to HEDN. If so, the second PAAM network seems useless.

Answer: Thanks for your comments. We've added a comparison of mechanical properties of HEDN1st and HEDN with the same polymer volume fraction. As shown in Fig. S16a, HEDN is much stronger than HEDN1st despite the same polymer volume fraction, indicating that the second PAAM network plays a critical role.

5. In Figure S7 and S8, it illustrated the gels show strain stiffening, what's the mechanism of the strain stiffening? Why there is an optimal value to HEDN-0.8? Higher swelling of HEDN1st should be adsorbed more second network monomer and should be shown higher entanglement, why the strain-stiffening capability is decreased after HED1st-0.8?

Answer: Thanks for your comments. Since HEDN1st network has higher effective cross-linking than the second PAAm network, the first network bears most of the external force at small strains, and the mechanical properties of HEDN are basically consistent with HEDN1st, showing a lower modulus. While at large strains, the second network begins to bear force, and the synergistic effect of the two networks highly orients the polymer chains and strengthens the hydrogel. It can be seen from Figure 4c that the birefringence of HEDN hydrogel increases dramatically at large strains, indicating that the polymer chains are highly ordered along the stretching direction, thus a stiffened material. The formation of this highly oriented structure is based on sliding entanglements and the cooperation of the two networks.

Higher swelling rates do absorb more second network monomers, but this does not mean more entanglements. As the first network swells, its number of entangled points per unit volume will decrease, but the large number of second networks can form new entanglements. Therefore, the total entanglement of HEDN hydrogels with different swelling ratios is jointly controlled by the above two effects. As can be seen from Figure S3 (blue area), the change in the swelling rate (determined by the AMPS content) of HEDN1st does not affect the entanglement of HEDN hydrogel. The strain-stiffening capability of HEDN depends on the ratio of the two networks. Too much or too little of one network will have adverse effects.

6. The authors have the wrong understanding of “homogeneous network”. The SEM and SAXS were used to interpret the homogeneous network. Up to date, only the tetra-PEG hydrogels are considered as near-homogeneous network. Also, what's the meaning of “homogenous orientation”? The concept is unclear.

Answer: We fully agree with you. Tetra-arm PEG hydrogels are considered as near-homogeneous network, because the molecular chain lengths between each crosslinking point are almost exactly the same. SEM and SAXS also cannot observe this small-scale structure. In this work, the definition of “homogeneous network” refers to whether there are micro-domains with different cross-linking densities. The microdomain is a local polymer network with the same cross-linking degree. And, these microdomains have larger sizes than molecular chains that can be detected by SEM and SAXS. As shown in Figure 3a, the areas marked by orange circles are less cross-linked, and the areas marked by blue circles are more cross-linked, representing the two types of microdomains respectively. When this microdomain difference is reduced, or there are no microdomains, we call it a more uniform network. The manuscript has used “uniform” instead of “homogeneous” to avoid confusion.

Due to the difference of crosslinking density, the two microdomains will have different deformations under strain, so the orientation of the network is not uniform. HEDN hydrogels have a more uniform orientation structure by sliding entanglement. Relevant conceptual explanations have been added to the revised manuscript. (See the first paragraph on Page 14 in the revised manuscript)

Revisions:

Because the hydrogel network has inhomogeneous cross-linked microdomains, the deformation of these microdomains is different when stretched, so the orientation of the network is not completely uniform. ...

7. In Figure 4d, it's clearly found there is a peak of 0.6 nm^{-1} (correlation length of $\sim 10.46 \text{ nm}$) in HEDN-0.8, indicating there is chain aggregation in the HEDN. The aggregation indicates the HEDN isn't a homogenous network. And, why is there chain aggregation? The scale of chain entanglement cannot explain such large aggregation.

Answer: Thanks for your comments. We believe that the cause of the peaks is the non-uniform distribution of polymer chains in the HEDN, that is, the formation of

microdomains with different entanglement density. Similar to TDN hydrogels, dense physical entanglement can also cause local differences in the cross-linking density of networks, resulting in the formation of microdomains. Unlike the chemical crosslinking points, the sliding entanglements can adjust the network when swelling, and the differences in the microdomain are weakened.

8. Also, in Figure 4d and 4f, the intensity after stretching increased, why?

Answer: Almeida et al. believed that large differences in SAXS scattering profiles at low-q regions were attributable to changes in network architecture⁶. Rowan et al. used SAXS to study the structural changes in the strain stiffening behavior of hydrogels, and found that the scattering intensity in the low q region increased with stress⁷. They also believe that this is a change in the network structure at larger length scales, and that the deeper cause is the anisotropy caused by the pores in the network being stretched along the direction of the stress. Compared with the non-uniform stretching of the TDN hydrogel network, due to the whole HEDN network is significantly stretched, the non-uniform entanglement network becomes a highly uniform anisotropic orientation structure, which is the reason for the increase in scattering intensity.

9. The HEDN gel demonstrated good recovery with low hysteresis. Therefore, I think the peak should be found again after unloading.

Answer: Thank you for your constructive comments. Yes, as you expected, the peak in HEDN gel reappears after unloading (Fig. S20). This data has been added to the manuscript to further characterize the good recovery of HEDN hydrogels. (See the first paragraph on Page 15 in the revised manuscript)

Revisions:

... SAXS profiles and patterns also indicate that HEDN hydrogels have high

recoverability (Fig. S20). The SAXS profiles before and after stretching show almost exactly overlapping curves, indicating that the HEDN hydrogels evolved from a non-uniform initial state to a uniform network structure after stretching, and then returned to a non-uniform structure after unloading. This recovery from a highly ordered stretch structure to a disordered entanglement structure is driven by entropy. ...

Fig. S20 **a** One-dimensional SAXS profiles of HEDN-0.8 hydrogel. **b-c** 1D SAXS profiles of HEDN-0.8 hydrogels with different strains in parallel (**b**) and perpendicular (**c**) directions. **d** 2D SAXS patterns of HEDN-0.8 hydrogels with different strains.

10. Compared to HEDN1st with a fatigue threshold of ~ 200 J/m², can this DN structure improve the fatigue threshold?

Answer: Yes. We have preliminary found that the DN structure can further improve the fatigue threshold of highly entangled hydrogels. As shown in the figure below, the fatigue threshold of 880 J/m² of HEDN hydrogel is significantly improved compared to HEDN1st of 160 J/m². Since the scope of this paper is the unification of high toughness and low hysteresis, fatigue resistance of other hydrogels based on this strategy and mechanism will be reported in another work.

Figure. Crack extension per cycle dc/dN versus applied energy release rate G for HEDN1st-0.8 and HEDN-0.8 hydrogels

References

1. Sakai, Takamasa, et al. "Design and fabrication of a high-strength hydrogel with ideally homogeneous network structure from tetrahedron-like macromonomers." *Macromolecules* 41.14 (2008): 5379-5384.
2. Di Lorenzo, F., and S. Seiffert. "Nanostructural heterogeneity in polymer networks and gels." *Polymer Chemistry* 6.31 (2015): 5515-5528.
3. Seiffert, Sebastian. "Scattering perspectives on nanostructural inhomogeneity in polymer network gels." *Progress in Polymer Science* 66 (2017): 1-21.
4. Tanaka Y, Kuwabara R, Na Y H, et al. Determination of fracture energy of high strength double network hydrogels[J]. *The Journal of Physical Chemistry B*, 2005, 109(23): 11559-11562.
5. Furukawa H, Kuwabara R, Tanaka Y, et al. Tear velocity dependence of high-strength double network gels in comparison with fast and slow relaxation modes observed by scanning microscopic light scattering[J]. *Macromolecules*, 2008, 41(19): 7173-7178.
6. De Almeida P, Jaspers M, Vaessen S, et al. Cytoskeletal stiffening in synthetic hydrogels[J]. *Nature communications*, 2019, 10(1): 609.
7. Jaspers M. Mechanics and Structure of Strain Stiffening Biomimetic Hydrogels[D]. [SI]:[Sn], 2017.

REVIEWER COMMENTS

Reviewer #1 (Remarks to the Author):

This version is greatly improved and can be considered acceptable.

Reviewer #2 (Remarks to the Author):

In this revised manuscript, Wang and coworkers reported tough double network hydrogels with low hysteresis based on highly entangled networks. Although the enhanced strength and low hysteresis of the gels were observed, the concept of DN gel with low hysteresis and strain-stiffening, emphasized on the innovation in this work, had been reported by Weiss and coworker (*Macromolecules*, 2016, 49, 8980-8987, Figure 7b in their work). I also don't think the novelty mentioned in the revised manuscript is high enough to be published by Nature Comm. Moreover, I agree with the second reviewer that there are still many misleading and indefinable concepts and interprets in this manuscript. There are also some comments about this work:

- (1) In this work, some concepts, such as rapid self-reinforcement, highly entangled, free sliding, highly uniform oriented, and so on, are indefinable. The concepts of non-uniform network and uniform network based on SEM are also misleading.
- (2) To increase the osmotic pressure of the polymer network, AMPS is used as a comonomer. Therefore, the HEDN1st can adsorb more second AAm monomer into the network. However, osmotic pressure will contribute to the elasticity. The usage of stiffness to evaluate the entanglements in the gel is incorrect. In Suo's work (*Science*, 2012, 374, 212), this method is valid because there is no osmotic pressure and it's extremely low cross-linker used. As a result, the high entanglements in the first network don't have the evidence. In the response letter and main text in revised manuscript, the authors stated the swell hydrogels have less entanglement per unit volume. Therefore, HEDN1st with more AMPS will show lower entanglements. How can the author to define the entanglements is high?
- (3) To improve the entanglements, the cross-linker content in second PAAm network should be also small, why such a high cross-linker of 0.05 mol% was needed?
- (4) In Figure 1d and Figure S16, the authors compared the gels with the same water content, however, the author didn't mention how can they control the gel with the same water content.
- (5) The free sliding between chains is also lack of evidence.

Point-by-point response to the reviewers' comments

Reviewer #1 (Remarks to the Author):

This version is greatly improved and can be considered acceptable.

Answer: We are very grateful for the reviewer's positive comments.

Reviewer #2 (Remarks to the Author):

In this revised manuscript, Wang and coworkers reported tough double network hydrogels with low hysteresis based on highly entangled networks. Although the enhanced strength and low hysteresis of the gels were observed, the concept of DN gel with low hysteresis and strain-stiffening, emphasized on the innovation in this work, had been reported by Weiss and coworker (Macromolecules, 2016, 49, 8980-8987, Figure 7b in their work). I also don't think the novelty mentioned in the revised manuscript is high enough to be published by Nature Comm. Moreover, I agree with the second reviewer that there are still many misleading and indefinable concepts and interprets in this manuscript. There are also some comments about this work:

Answer: Dear reviewer, we sincerely thank you for your time and effort in reviewing our paper again. We have revised the manuscript and added new data after carefully considering your comments and suggestions. All modifications are shown in red in the revised manuscript.

The work reported by Weiss and coworker mainly focuses on the research of single networks, double networks, triple networks, and quadruple networks composed of loosely cross-linked networks. This work mainly studies the effect of sliding entanglement on the structure and properties of DN hydrogels. Therefore, there is a significant difference in the research content between the two works. The DN gel mentioned by Weiss belongs to the elastic gel that is recognized as fragile and notch-sensitive. Although this DN network has low hysteresis and strain-stiffening

phenomena, the fracture energy, the most important factor restricting the application of hydrogels, has not been studied.

We resolve the toughness-recovery conflict, in addition to achieving ultra-high strain-stiffening capability, by fabricating an unusual double-network polymer in which entanglements function as slip links in the first network. The slide-ring PEG hydrogels with strain-induced crystallization prepared by Kohzo Ito and coworkers also achieved high toughness and low hysteresis, this work was published in “*Science*” (Science 372, 1078–1081 (2021)). High-toughness and low-hysteresis hydrogels prepared by Feng Yan and coworkers based on nanoconfined polymerization strategy were published in “*Nat. Mater.*” (10.1038/s41563-023-01697-9) and “*Angew. Chem. Int. Ed.*” (Angew. Chem. Int. Ed. 2023, e202316375). This work does not introduce any complicated functional molecules, and higher toughness and near-zero-hysteresis can be achieved at higher water content (89% in this work, 70%, 67% in the above work) by changing the topology of the gel. Thus, the method provided in this work is easier and more effective. These performance improvements are based on a new toughening mechanism, strain-induced highly uniform orientation based on slip links.

Given the effectiveness of this method and the detailed study of the unreported mechanism, we think this work is suited for publication in “*Nature Communications*”.

Misleading and indefinable concepts and interpretations have been carefully revised.

(1) In this work, some concepts, such as rapid self-reinforcement, highly entangled, free sliding, highly uniform oriented, and so on, are indefinable. The concepts of non-uniform network and uniform network based on SEM are also misleading.

Answer: Thank you for your careful comments. We have defined the above concepts accurately and checked the full text. We also agree that some SEM may be misleading to readers, therefore, we have added an accurate note to the legend in Fig. 3. Free sliding is a mistake in our expression. We only want to express that polymer chains can slide between each other. We have corrected this description.

Revisions:

... The higher strain-stiffening capacity allows soft materials to harden more rapidly under large deformation, which is defined as rapid self-reinforcement behavior.

Highly entangled hydrogels mean that there is more entanglement per unit volume under the same solid content. ...

... If when the network is stretched and the microdomains disappear, the resulting oriented structure is regarded as highly uniformly oriented. ...

... Note: This image indicates a more uniform network compared to Fig. 3a, but there are still microdomains that only can be detected by SAXS.

(2) To increase the osmotic pressure of the polymer network, AMPS is used as a comonomer. Therefore, the HEDN1st can adsorb more second AAm monomer into the network. However, osmotic pressure will contribute to the elasticity. The usage of stiffness to evaluate the entanglements in the gel is incorrect. In Suo's work (Science, 2021, 374, 212), this method is valid because there is no osmotic pressure and it's extremely low cross-linker used. As a result, the high entanglements in the first network don't have the evidence. In the response letter and main text in revised manuscript, the authors stated the swell hydrogels have less entanglement per unit volume. Therefore, HEDN1st with more AMPS will show lower entanglements. How can the author to define the entanglements is high?

Answer: Thank you for your constructive comments. The stiffness to evaluate the entanglements in the gel is also valid.

Firstly, we hypothesize that osmotic pressure affects network elasticity in this system. HEDN1st-0.8-W_x (x=2, 4, 6, 8, 10) gels prepared at different monomer concentrations have the same solid content and the same AMPS content, so the elasticity caused by osmotic pressure is equivalent in these gels. However, the stiffness of these gels is not equal (Fig. S4). The difference in stiffness shown in Fig. S4 should be attributed to entanglements, with greater stiffness indicating more entanglements.

Secondly, in the salt solution, the osmotic pressure is suppressed because of the Donnan equilibrium¹. Therefore, we prepared hydrogels swollen by salt solutions (3 M NaCl) to eliminate the effect of osmotic pressure on elasticity. As shown in Fig. S3, there is no significant difference in the stress-strain curves and stiffness of hydrogels swollen by water or salt solution, whether at low AMPS content, such as HEDN1st-0.2, or at higher AMPS content, such as HEDN1st-2.0, which means that the elasticity caused by osmotic pressure is negligible in this system. The reason may be that the elasticity caused by polymer chains is much higher than the osmotic pressure at this polymer fraction. We have added this data in the revised manuscript to avoid misleading readers.

Finally, the equilibrium water content of the hydrogel can also be used as evidence of chain entanglement (Fig. S1). Increasing concentration is a common method to increase chain entanglement. The large differences in water content of these gels, which were prepared with the same chemical composition but at different monomer concentrations, can be attributed to chain entanglement.

In this work, unless otherwise specified, high entanglement refers to the gel with a higher entanglement density at the same polymer fraction. Before comparing the entanglement degree, the hydrogels must be processed to the same solids content.

Fig. S3 Tensile stress-strain curves of HEDN1st-0.2 (a), HEDN1st-0.8 (b), HEDN1st-1.4 (c) and HEDN1st-2.0 (d) hydrogels swollen by water or salt solutions (3 M NaCl).

The stiffness of these hydrogels (e). Polymer fraction is 11%.

Revisions:

...To eliminate the effect of osmotic pressure on stiffness, we prepared hydrogels swollen by salt solutions (3 M NaCl). As shown in Fig. S3, there is no significant difference in the stress-strain curves and stiffness of hydrogels swollen by water or salt solution, whether at low AMPS content, such as HEDN1st-0.2, or at higher AMPS content, such as HEDN1st-2.0, which means that the elasticity caused by osmotic pressure is negligible in this system. The HEDN1st hydrogels obtained at five different concentrations were processed into samples with a water content of 89% to eliminate the influence of polymer contents on stiffness. Therefore, the stiffness of the gel is determined by the degree of entanglement in this work. ...

(3) To improve the entanglements, the cross-linker content in second PAAm network should be also small, why such a high cross-linker of 0.05 mol% was needed?

Answer: Thanks for your comments. When the first network is stretched, the second network requires a specific cross-link density to achieve the synergistic effect of the two networks. If the second network is too loose, the first network will break before the second network is loaded. On the contrary, too high cross-link density of the second network will make the gel brittle.

(4) In Figure 1d and Figure S16, the authors compared the gels with the same water content, however, the author didn't mention how can they control the gel with the same water content.

Answer: Thanks for your precious comments. Yes, we should make this important point clearer. We have added this method in the Methods section.

Revisions:

Preparation of gels with specific water content. A quantitative amount of hydrogel

with known water content or completely dehydrated gel is placed in a sealed box. After adding an appropriate amount of water, it is placed in a constant-temperature oscillator at 37 °C and incubated for 48 hours to obtain a gel with a determined water content.

(5) The free sliding between chains is also lack of evidence.

Answer: Thank you very much for your careful review. This comment leads us to discover that free sliding is a misnomer. We only want to express that polymer chains can slide between each other. We have corrected this misnomer. It is also known that entanglements can slip. As described in these works:

“Entanglements function as slip links...” (Science, 2021, 374, 212).

“As entangled chains are ‘mobile’ in the network...” (Nature, 618, 740–747 (2023)).

“Entanglements in swollen hydrogels readily slip and negligibly dissipate energy before rupture.” (Adv. Mater. 2022, 2206577).

“We employ a novel methodology, which analyzes entanglement constraints into a complete set of pairwise interactions, similar to slip links.” (Macromolecules 2012, 45, 9475–9492).

Revisions:

This **sliding** entanglement...

The **sliding** entanglement points...

1. Ricka, J.; Tanaka, T. Swelling of ionic gels: quantitative performance of the Donnan theory. *Macromolecules* 1984, 17, 2916-2921.

REVIEWERS' COMMENTS

Reviewer #2 (Remarks to the Author):

The authors have addressed the comments from the reviewers, and I think it can be accepted in current state.